# Tumor Immune Microenvironment Biomarkers for Recurrence Prediction in Locally Advanced Rectal Cancer Patients after Neoadjuvant Chemoradiotherapy

**DOI:** 10.3390/cancers16193353

**Published:** 2024-09-30

**Authors:** Jun-Eul Hwang, Sung-Sun Kim, Hyun-Jin Bang, Hyeon-Jong Kim, Hyun-Jeong Shim, Woo-Kyun Bae, Ik-Joo Chung, Eun-Gene Sun, Taebum Lee, Chan-Young Ock, Jeong-Seok Nam, Sang-Hee Cho

**Affiliations:** 1Division of Hematology and Oncology, Department of Internal Medicine, Chonnam National University Medical School and Hwasun Hospital, Hwasun 61469, Republic of Korea; juneul2@gmail.com (J.-E.H.); daunjin@gmail.com (H.-J.B.); gohjkim@naver.com (H.-J.K.); hjnhj@chonnam.ac.kr (H.-J.S.); drwookyun@chonnam.ac.kr (W.-K.B.); ijchung@jnu.ac.kr (I.-J.C.); egsun@jnu.ac.kr (E.-G.S.); 2Department of Pathology, Chonnam National University Medical School, Gwangju 61469, Republic of Korea; kimsspathology@jnu.ac.kr; 3National Immunotherapy Innovation Center, Chonnam National University Medical School, Hwasun 61469, Republic of Korea; 4Lunit, Seoul 06159, Republic of Korea; taebum.lee@lunit.io (T.L.); ock.chanyoung@lunit.io (C.-Y.O.); 5School of Life Sciences, Gwangju Institute of Science and Technology, Gwangju 61005, Republic of Korea; namje@gist.ac.kr

**Keywords:** tumor microenvironment, rectal neoplasms, neoadjuvant therapy, biomarkers, artificial intelligence

## Abstract

**Simple Summary:**

This study aimed to identify prognostic factors by combining clinicopathologic parameters with tumor microenvironment (TME) biomarkers in patients with locally advanced rectal cancer (LARC) who underwent surgery following neoadjuvant chemoradiotherapy (nCRT). We analyzed CD8+ T cells, CXCR3, CXCL10, and α-SMA using immunohistochemical staining and incorporated AI-powered digital pathology to assess the spatial TME. Our findings showed that high expression of CD8+ T cells, CXCR3 in tumor-infiltrating lymphocytes (TILs), and an inflamed phenotype were associated with better recurrence-free survival (RFS). However, these factors were not predictive of overall survival (OS). Patients with an immune-desert phenotype had a poor prognosis regardless of pathologic stage or the administration of postoperative chemotherapy. These results suggest that CD8+ T cells and AI-powered immune phenotypes, together with clinical factors, can guide personalized treatment strategies in LARC patients post-nCRT and highlight the potential benefits of modifying the tumor immune microenvironment (TiME) to reduce recurrence after surgery.

**Abstract:**

Background/Objectives: The tumor microenvironment (TME) has emerged as a significant prognostic factor. This study aimed to identify prognostic factors by combining clinicopathologic parameters and the TME biomarkers in patients who underwent surgery following neoadjuvant chemoradiotherapy (nCRT) for locally advanced rectal cancer (LARC). Methods: CD8^+^ T cells, CXCR3, CXCL10, and α-smooth muscle actin (α-SMA) were analyzed via immunohistochemical staining. We also incorporated AI-powered digital pathology to assess the spatial TME. The associations between these biomarkers, clinicopathologic parameters, and survival outcomes were evaluated. Results: CD8^+^ T cell expression, CXCR3 expression in tumor-infiltrating lymphocytes (TILs), and immune phenotypes were correlated. LARC patients with a high expression of CD8^+^ T cells, CXCR3 in TILs, and an inflamed phenotype had a significantly better prognosis than their counterparts did. In the multivariate analysis, the expression of CD8^+^ T cells and the inflamed/immune-excluded phenotype were significant tumor immune microenvironment (TiME) biomarkers for recurrence-free survival (RFS) but not for overall survival (OS). Notably, patients with the immune-desert phenotype had a poor prognosis regardless of pathologic stage, even if postoperative chemotherapy was administered (*p* < 0.001). Conclusions: CD8^+^ T cells and AI-powered immune phenotypes, alongside clinical factors, can guide personalized treatment in LARC patients receiving nCRT. A therapeutic strategy to modify the TiME after nCRT could help reduce recurrence after surgery.

## 1. Introduction

Unlike colon cancer, the standard treatment for locally advanced rectal cancer (LARC) includes neoadjuvant chemoradiotherapy (nCRT) before surgery to reduce local recurrence and increase the organ preservation rate [1,2]. Recently, increasing pathological complete response (pCR) has been attempted by adding total neoadjuvant therapy (TNT) to nCRT. On the basis of these results, further studies are being conducted to explore radiation-sparing approaches or de-escalation strategies according to the tumor response to avoid adverse effects [3].

However, even while receiving the same treatment at the same stage, not all patients can reach pCR after preoperative treatment because tumor responses can vary from person to person owing to tumor heterogeneity. Thus, postoperative chemotherapy is administered to patients with remnant tumors or a poor response after nCRT. The most important parameter for selecting patients requiring postoperative chemotherapy is the pathologic stage (yp stage), such as high-risk stage II or III, including the tumor regression grade (TRG), which represents the response to preoperative treatment [4]. Although standard adjuvant (postoperative) chemotherapy, such as 5-FU monotherapy or oxaliplatin chemotherapy, is performed after surgery, recurrence remains a major concern. Thus, an innovative therapeutic strategy beyond current standard chemotherapy is urgently needed to overcome treatment outcomes. Furthermore, there are still unmet needs to detect patients at high risk of recurrence, sometimes leading to overtreatment in patients who do not need adjuvant therapy.

From these perspectives, the exploration of novel, specific and accurate tumor biomarkers has advanced. Additionally, the focus of tumor biomarkers is shifting not only to the tumor itself but also to the tumor microenvironment (TME). The TME is a highly complex ecosystem in which tumor cells coexist with immune cells and nonimmune cells, such as endothelial cells and stromal cells. The tumor immune microenvironment (TiME), especially CD3^+^ and CD8^+^ T cells, is a critical component of colorectal cancer (CRC) progression [5]. Previous studies have shown the potential role of immune infiltrates in the prediction of radio-responsiveness to nCRT in rectal cancer [6]. Particularly, CXCL10 is a chemokine that plays a critical role in attracting immune cells, including CD8+ T cells, to sites of inflammation or tumors. CXCR3, a receptor expressed on the surface of activated CD8 T cells, binds to CXCL10 and other related chemokines. This interaction facilitates the migration of CD8+ T cells to tumor, which can enhance the immune response in various pathological condition [7]. In addition to the TiME, the stroma produces and secretes growth factors, cytokines, and chemokines that promote tumor progression and immune evasion. Fibroblasts are the dominant component of the tumor stroma, with cancer-associated fibroblasts (CAFs), which are activated by tumor cells, playing a key role in the TME, involving cancer-promoting or cancer-restraining effects and metastasis [8,9,10]. Verset, L. et al. revealed the prognostic significance of the CAF marker using a ratio of α-smooth muscle actin (α-SMA)/epithelial area associated with decreased recurrence-free survival (RFS) in LARC patients [11].

However, the interpretation of rectal cancer requires caution because nCRT not only directly kills tumor cells but can also modify the immune landscape. The release of tumor antigens from dying cancer cells after nCRT can stimulate an antitumor immune response by increasing cytokine levels and attracting immune cells to the tumor site (the antitumor effect). In contrast, it can also induce the expression of immunosuppressive factors, which might protect the remaining tumor cells. Most previous studies revealed biomarkers of the TME in tissues at diagnosis to predict the response to nCRT [6,12]. However, analyzing tissue after nCRT might be more accurate than analyzing tissue at diagnosis because tumor heterogeneity affects response. Additionally, identifying biomarkers through a comprehensive analysis of clinical factors alongside the TME is crucial for improving prognosis prediction.

Therefore, this study aimed to identify optimal biomarkers associated with the TiME, stroma and clinicopathologic data through comprehensive analysis via a multifaceted approach, including artificial intelligence (AI)-based digital pathology in LARC patients following nCRT.

## 2. Materials and Methods

### 2.1. Patients and Samples

A total of 192 consecutive LARC patients treated with nCRT at Chonnam National University Hwasun Hospital between January 2015 and December 2019 were reviewed. Of these, 108 patients were included in this retrospective study (Appendix A).

The inclusion criteria were (1) histologic diagnosis of primary rectal adenocarcinoma, (2) clinical stage III at diagnosis, (3) absence of synchronous malignancy, (4) staging workup including rectal MRI, (5) completion of long-course nCRT with fluorouracil-based chemotherapy, (6) complete information of clinical/pathologic stages/adjuvant treatment and survival and (7) achievable tissue samples from surgical resection specimens after nCRT in the Biobank of Chonnam National University Hwasun Hospital (CNUHH), a member of the Korea Biobank Network, with informed consent. We excluded patients with microscopic or grossly noncurative resection, inadequate lymph node dissection (<12 lymph node dissections), synchronous malignancies and <30 days of premortality hospitalization, as well as those diagnosed with other malignancies during follow-up. The institutional review board (CNUHH-2021-005) and the data review board (CNUHH-D2023-004) at CNUHH approved this study.

### 2.2. Staging, Treatment, and Follow-Up

The staging workup for rectal cancer comprises digital rectal examination; endoscopy with biopsy for histologic diagnosis; computed tomography (CT) scans of the abdomen, pelvis, and chest; and rectal magnetic resonance imaging (MRI). The eighth edition of the American Joint Committee on Cancer TNM staging system was used to assess clinical tumor stages. For ambiguous lesions, positron emission tomography (PET)/CT was performed.

After being diagnosed with stage III rectal cancer, all patients underwent nCRT. During nCRT, a total radiation dose of 50.4 Gy was applied in 28 fractions of 1.8 Gy. Additionally, oral capecitabine (825 mg/m^2^ twice daily during the 5-plus-week course of radiotherapy) was concurrently administered. After 6–8 weeks, the treatment response was assessed, and total mesorectal excision was performed to surgically remove the residual tumor. Adjuvant chemotherapy was subsequently administered according to the physician’s decision on the basis of the surgical pathology and the patient’s general condition.

After completing adjuvant chemotherapy, physical examination, CT images, and serial serum CEA levels were checked every 3–6 months for 5 years. Colonoscopy was performed every 1–3 years depending on risk factors. For those who were lost to surveillance, official survival data were obtained from the Ministry of the Interior and Safety of the Republic of Korea, thus suggesting the completeness of survival status in the entire study population.

### 2.3. Pathologic Tumor Stage and Tumor Regression Grading

Surgical tissue samples from surgery patients were microscopically examined to evaluate tumor responses to chemoradiotherapy. The eighth edition of the American Joint Committee on Cancer TNM staging system was used to assess pathologic tumor stages.

Tumor regression grading (TRG) of the primary tumor treated with nCRT was semiquantitatively evaluated on hematoxylin and eosin (H&E)-stained slides according to modified Dworak’s criteria as previously described [13]. The characteristics of each grade were as follows: TRG 1, predominant tumor cell mass (>50%) with obvious fibrosis or no regression; TRG 2, dominantly fibrotic changes with few tumor cells or groups; TRG 3, very few tumor cells (one or two microscopic foci < 0.5 cm in diameter); and TRG 4, no tumor cells.

### 2.4. Immunohistochemical Staining

Two to three representative foci of the tumor and one focus of the matched non-neoplastic mucosal tissue and metastatic lymph node (if applicable) in each case were selected for TMA blocks. Tissue cores (2 mm in diameter) were obtained from each paraffin block to make a TMA block. Immunohistochemistry (IHC) was performed on the TMA slides via an autostainer (Bond-Max automated IHC/ISH stainer; Leica Biosystems, Wetzlar, Germany) with primary antibodies against the following proteins: α-SMA (dilution 1:200; catalog no. 19245, Cell Signaling Technology, Danvers, MA, USA), CXCR3 (dilution 1:2000, catalog no. 288437, Abcam, Cambridge, UK), CXCL10 (dilution 1:1000, catalog no. 306587, Abcam, Cambridge, UK), and CD8 alpha (dilution 1:1000, catalog no. 237709, Abcam, Cambridge, UK). The immunoreactivity for α-SMA was categorized as mild or moderate and marked by the value multiplied by the intensity and area. The immunoreactivity for CXCR3 in TME was interpreted as 1, ≤33% of positively reactive lymphoid cells; 2, 34–67% of positive cells; and 3, >67% of positive cells. CD8 alpha immunopositivity was interpreted as 1, <10% of positive cells; 2, 10–30% of positive cells; and 3, >30% of positive cells. CXCL10 positivity was categorized as 0, no positive cells; 1, <5% positive cells; and 2, >5% positively reacting cells. We identified TILs as the lymphocytes located in the stroma surrounding the tumor in H&E and IHC slides. The evaluation of expression was performed manually by an expert pathologist.

### 2.5. Neoadjuvant Rectal Score

The neoadjuvant rectal score (NAR) assesses the difference between the initial clinical and pathological T stage and the presence or absence of nodal involvement after treatment [14]. The calculation formula incorporates cT-, ypT-, and ypN-stage information, applying discrete weighting values for each staging category as follows. The equation for calculating the NAR is as follows:$$NAR={\left[5ypN-3\left(cT–ypT\right)+12\right]}^{2}/9.61$$
where cT represents the clinical T stage (1, 2, 3 or 4), ypT represents the pathological T stage (1, 2, 3 or 4) and ypN represents the pathological nodal status (0, 1 or 2). In the present study, the score values of this population were categorized as low (<8), intermediate (8–16) and high (>16), following the validation results of the NSABP-03 and CAO/ARO/AIO-04 trials [14,15].

### 2.6. Neutrophil-to-Lymphocyte Ratio

The neutrophil-to-lymphocyte ratio (NLR) was calculated by dividing the percentage of neutrophils by the percentage of lymphocytes from the peripheral white blood cell count. The differential blood counts before surgery were extracted from the patients’ electronic medical records. An NLR of ≥5 was considered elevated according to an earlier report [16,17,18]. The cut-off values for the groups were defined by successively comparing different NLR values for their impact on the prognostic significance of RFS, dividing the two groups into low (<6) and high (≥6) NLRs.

### 2.7. AI-Based Immune Phenotype Analysis

Lunit SCOPE IO, an advanced AI-driven whole-slide image analyzer, was engineered utilizing data from over 26 distinct tumor types and origins. The cell detection model was constructed on the basis of 20,617 image patches sourced from 5609 whole-slide images (WSIs), with 3798 patches designated for training purposes and 1811 patches reserved for validation. The tissue segmentation model was developed via 76,110 patches extracted from 18,935 WSIs, with 15,936 patches allocated for training and 2999 patches allocated for validation. A cumulative total of 2,828,448 cells and 1.07 × 10^10^ μm^2^ of cancerous tissue and cancer stroma were annotated by board-certified pathologists.

Lunit SCOPE IO predicts cancer areas (cancer parenchyma), cancer stroma, and lymphocytes at the pixel level. It then identifies lymphocytes located in both the cancer area and cancer stroma as TILs, which are used to differentiate immune phenotypes. Spatial TIL densities were assessed by detecting cellular components and segmenting tissue regions within H&E-stained whole-slide images. Each WSI, varying in dimension, was partitioned into grids of 0.25 mm^2^ for meticulous analysis. The model estimated TIL densities and classified immune phenotypes on the basis of the following criteria: inflamed, characterized by an intratumoral TIL density ≥ 130/mm^2^; immune-excluded, defined as intratumoral TIL density < 130/mm^2^ with stromal TIL density ≥ 260/mm^2^; and immune-desert, where TIL densities fell below the specified thresholds in both regions. The representative immune phenotype for each WSI was classified as inflamed if ≥33.3% of the grids within the WSI exhibited the inflamed immune phenotype and immune-excluded if ≥33.3% of the grids displayed the immune-excluded phenotype and the proportion of inflamed phenotype was <33.3%; otherwise, the WSI was categorized as immune-desert.

### 2.8. Statistical Analysis

Analyses of associations among clinicopathological parameters were performed via the chi-square test and Fisher’s exact test. Survival analyses were performed via the Kaplan–Meier method, and curves were compared via the log-rank test. Overall survival (OS) was defined as the time from the date of surgery to the date of death. RFS was defined as the time from the date of surgery to the date of recurrence or death, whichever occurred first. If neither event occurred at the time of analysis, the patient was censored. Factors associated with OS and RFS were identified by univariate and multivariate Cox proportional hazard regression models with hazard ratios (HRs) and 95% confidence intervals (CIs). All variables from the univariate analysis with *p*-values < 0.05 were incorporated into the multivariate Cox hazard regression model with a stepwise forward procedure. Statistical analyses were performed via SPSS version 24.0 (IBM Corp., Armonk, NY, USA). All *p*-values were two-sided, with *p*-values < 0.05 indicating statistical significance.

## 3. Results

### 3.1. Patient Population, Clinical Characteristics and Survival Outcomes

A total of 108 patients with a median age of 60 years were analyzed (Table 1). The median follow-up was 59 months. During this time, 32 patients experienced recurrence, and 19 patients died due to disease recurrence. The 3-year RFS was 75.6% (95% CI: 67.17–84.03), and the 5-year OS was 84.8% (95% CI, 77.15–91.65). Adjuvant chemotherapy included oral capecitabine as 5-fluorouracil monotherapy (*n* = 41) or oxaliplatin-based combination therapy (*n* = 55). Seven patients did not receive adjuvant chemotherapy. Survival outcomes differed significantly based on whether adjuvant chemotherapy was administered (RFS: *p* < 0.001, OS: *p* < 0.001). However, outcomes did not differ on the basis of the type of adjuvant chemotherapy (i.e., 5-FU monotherapy or oxaliplatin-based combination therapy); hence, we categorized the groups as either “none” or “adjuvant chemotherapy”. Table 2 summarizes the pathologic results after surgery.

### 3.2. Associations of the Clinicopathologic Parameters, TRG, NAR, and NLR, with Survival Outcomes

Among these patients, tumor downstaging was observed in 47 patients (yp stage I = 11, yp stage II = 36), whereas the remaining patients remained at yp stage III at diagnosis, although there was improvement in the T and N stages. As expected, a lower yp stage was associated with a higher TRG (*p* = 0.004) and a lower NAR (*p* < 0.001). Patients with an NLR ≥ 6 had significantly poorer RFS than those with an NLR < 6. Additionally, a higher NLR in blood was significantly associated with lower CD8 expression in tumor tissues (*p* = 0.014).

### 3.3. Immunohistochemical Staining for CD8^+^ T Cells and CXCR3 in TILs for Predicting RFS

Grade 3+ of CD8^+^ T cells was observed in 40 patients (37%), whereas grade 1+ of CD8^+^ T cells was observed in 13 patients (12%; Supplement Appendix A). Survival analysis revealed that patients with grade 2+ and 3+ of CD8^+^ T cells had similar survival curves, but those with grade 1+ of CD8^+^ T cells were associated with significantly poorer RFS (*p* < 0.001, Appendix A). Therefore, we categorized the groups as having either a low expression of CD8^+^ T cells (grade 1+) or high expression of CD8^+^ T cells (grade 2+ and grade 3+). Patients in the low-expression group of CD8^+^ T cells had poorer RFS (3-year RFS rate of 23%) than those in the high-expression group of CD8^+^ T cells (3-year RFS rate of 81.3%), regardless of yp stage. We analyzed RFS only in the yp stage II patients (*n* = 80) to explore the role of CD8 expression within the same stage and found a significant difference in RFS between the two groups (*p* < 0.001).

In the case of CXCR3 in TILs, grade 3+ was observed in 10 patients (9%), whereas grade 1+ was observed in 55 patients (51%). Survival analysis revealed that patients with grade 1+ and grade 2+ CXCR3 had similar survival curves, but those with grade 3+ CXCR3 had better RFS (*p* = 0.066, Appendix A). On this basis, we categorized the groups as having either a low expression of CXCR3 (grade 1+, grade 2+) or high expression of CXCR3 (grade 3+). Similar to patients with CD8^+^ T cells, patients in the high-expression group of CXCR3 had significantly longer RFS than those in the low-expression group of CXCR3 (*p* = 0.023).

### 3.4. Correlation of the Immune-Desert Immune Phenotype with CD8^+^ T Cells and CXCR3 in TILs

We performed immune phenotype analysis via an AI-powered spatial TIL analyzer from H&E images of the corresponding tumor samples, to analyze the TME comprehensively. Only 7 patients (6%) exhibited an inflamed phenotype, whereas 36 patients (33%) presented an immune-desert phenotype. This immune phenotype was a significant predictor of RFS, with the following order of prognosis: inflamed > immune-excluded > immune-desert (*p* = 0.001, Appendix A). For the multivariate analysis related to RFS, we divided the groups into inflamed, immune-excluded, and immune-desert groups (*p* = 0.005, Appendix A). As depicted in Figure 1, the expression of CD8^+^ T cells, CXCR3 in TILs, and the immune phenotype were significantly correlated. Specifically, the expression level of CXCR3 tended to gradually increase in the order of immune-desert, immune-excluded, and inflamed phenotypes. This finding suggests a relationship in which CXCR3 expression levels may serve as a marker indicative of the immune status within the TME, influencing the observed immune phenotypes.

Unlike the expression of CD8^+^ T cells or CXCR3 in TILs, which was not correlated with TRG, immune phenotypes were significantly associated with TRG. Specifically, TRG 3 was observed in patients with an inflamed phenotype (85.7%), whereas TRG 1 was more prevalent among patients with an immune-desert phenotype (63.6%, *p* = 0.036, Figure 2). In addition, even within the same TRG 2 group, different RFS rates were observed on the basis of CD8^+^ T cell expression (*p* < 0.001*)* and immune phenotype (*p* = 0.037*)*.

### 3.5. Univariate and Multivariate Analysis of Clinical Factors and the Tumor Immune Microenvironment According to Survival

Among the significant risk factors identified in the univariate analysis of RFS, clinical factors, including tumor grade (G3), perineural invasion (PNI), circumferential resection margin (CRM) status, the administration of adjuvant chemotherapy and the expression of CD8^+^ T cells, CXCR3, and Lunit SCOPE IO as TiME biomarkers remained significantly associated with RFS in the multivariate analysis (Table 2). In contrast, while clinical factors such as PNI, CRM status, and the administration of adjuvant chemotherapy were significantly associated with OS in the multivariate analysis, TiME biomarkers did not demonstrate statistical significance in this study.

We conducted an additional stratified analysis based on adjuvant chemotherapy to assess whether the prognostic significance of these biomarkers regarding recurrence is influenced by adjuvant chemotherapy. As a result, even when adjuvant chemotherapy was administered, CD8^+^ T cells, CXCR3 in TILs and AI-powered immune phenotype analysis exhibited significant prognostic value for recurrence risk (Figure 3). Overall, we developed a predictive model combining yp stage and TiME biomarkers, including CD8^+^ T cells or AI-powered immune phenotype. Patients with a high expression of CD8^+^ T cells demonstrated better survival than other patients regardless of yp stage. In addition, patients with either an inflamed or immune-excluded phenotype and yp stage I–II disease had a significantly better 3-year RFS rates of 85.2% (95% CI:85.11–85.29) than patients without these phenotypes, who had a 3Y-RFS rate of 56.0% (95% CI:55.83–56.17). However, patients with the immune-desert phenotype had a poor prognosis regardless of yp stage (Figure 4).

## 4. Discussion

The relationship between the TiME and prognosis in patients with colon cancer has already been established in numerous studies [5,19,20]. However, the understanding of the complex causal links between the TiME and survival outcomes in rectal cancer patients, especially in the context of adjuvant chemotherapy, remains limited. Our study sheds new light on the clinical significance of the TiME in recurrence the prediction of LARC recurrence and highlights the potential role of novel digital biomarkers using AI-powered digital pathology to predict patient prognosis.

Although the ADORE trial established oxaliplatin-based combination chemotherapy as the most effective adjuvant chemotherapy for patients with yp stage II or III rectal cancer after nCRT, recurrence remains a significant issue [21]. Our findings, which are consistent with those of previous studies on colon cancer, underscore the importance of CD8^+^ T cells in predicting disease-free survival in rectal cancer patients, even after nCRT. Notably, the recurrence rate varies with CD8^+^ T cell expression, suggesting the need to improve the TiME to reduce recurrence in LARC patients.

Recent studies have revealed that the density of CD^3+^ and CD8^+^ T cells among immune cells in colon cancer is a good predictor of disease-free survival not only in microsatellite unstable but also in microsatellite stable colon cancer [22,23]. Thus, an immunoscore, which quantifies cytotoxic T cells in the TME by combining CD^3+^ and CD8^+^ T cell densities, was incorporated into the 2020 ESMO Clinical Practice Guidelines for localized colon cancer to refine prognosis in conjunction with TNM staging [24]. By validating the immunoscore in a large-scale clinical trial, Sinicrope, F.A. et al. demonstrated that it could enhance prognostication beyond clinical risk group classification in patients treated with adjuvant 5-FU and oxaliplatin chemotherapy in stage III colon cancer patients in the NCCCTG NO147 clinical trial [25]. Our results are consistent with this report, highlighting the importance of CD8^+^ T cells in patients with rectal cancer after nCRT. In particular, the variance in the recurrence rate, which depends on the expression of CD8^+^ T cells present even at the same pathologic stage, suggests that improvement of the TiME is necessary as an adjuvant treatment.

For CD8^+^ T cells to effectively infiltrate the tumor site, chemokine receptors, such as CXCR3, must interact with their corresponding chemokines (CXCL9, CXCL10 and CXCL11). CXCR3 binds and traffics toward its IFNγ-inducible ligands, CXCL9, 10, and 11, which are expressed primarily in activated CD8^+^ T cells, NK cells, and CD^4+^ T_H1_ cells and play critical roles in recruiting and retaining T cells during infection, autoimmunity and cancer [26]. Previous research has shown that CXCR3 is crucial for the efficacy of adoptively transferred antitumor T cells and the mediation of tumor regression following anti-PD-1 therapy. Radiation therapy can enhance CXCR3-mediated T cell trafficking through IFN-induced chemokine stimulation [27,28,29]. However, only a small percentage of patients (strong positivity in only 9% of patients in our study) exhibited high CXCR3 expression after nCRT, indicating that other factors, such as the TME composition, may regulate CXCR3. For example, TGF-β, a tumor-promoting cytokine, can block T cell trafficking by repressing CXCR3 expression [30]. Thus, modulating the TiME, including suppressing TGF-β, may be necessary for recruiting CD8^+^ or CXCR3^+^ T cells to promote the antitumor TME. Our multivariate analysis revealed that CD8^+^ T cells are powerful and definitive biomarkers that serve as key effector cells in various TME conditions. Accordingly, therapeutic approaches enhancing CD8^+^ T cell expression, such as promoting CXCR3 expression or inhibiting TGF-β, are needed. A current clinical trial using LY2157299 (a TGF-βR inhibitor) in combination with chemotherapy and radiotherapy may provide further insights into this hypothesis.

A major challenge in leveraging the TiME in clinical practice is the issue of spatial or temporal heterogeneity. For example, the CXCR3-CXCL9-CXCL-10-CXCL-11 axis has different effects on immune cells and tumor cells, and higher CXCR3 expression on tumor cells, unlike that on immune cells, is associated with a worse prognosis [20,31,32]. Therefore, we analyzed CXCR3 only in TILs to avoid conflicting interpretations. As a result, we found that CXCR3 expression on TILs, together with that on CD8^+^ T cells, serves as a positive prognostic factor for RFS. To the best of our knowledge, this is the first report on the prognostic role of CXCR3 in LARC patients.

As another approach to understanding the spatial TME and validating immune biomarkers, we introduced the Lunit Score IO in this study. CD8^+^ T cell densities were significantly correlated with the immune phenotype, reflecting the TiME. Prior studies have demonstrated that computational TIL analysis of H&E-stained images is correlated with patient prognosis in non-ICI-treated cancer patients. H&E-based immune phenotyping, which reflects active antitumor immune responses, correlates with high IFNG pathway activation. Computational TIL assessment offers a more objective, time-efficient, and labor-saving analysis, minimizing interobserver variability and interpretation bias [33,34,35]. Additionally, adjuvant therapy might not be necessary for inflamed or immune-excluded patients with yp stage I–II disease. Furthermore, inflamed, immune-excluded and immune-desert phenotypes were strongly correlated with a decrease in CXCR3 expression. Thus, in addition to CD8, CXCR3 has potential as a prognostic marker for the TiME, and further validation studies for CXCR3 are needed in colorectal cancer.

The ultimate goal of understanding the TiME is to apply this knowledge to therapeutic strategies. Our findings demonstrated that patients with low CD8^+^ T cell expression have a relatively high recurrence rate even with oxaliplatin-based chemotherapy, indicating that the use of cytotoxic chemotherapy alone is insufficient to overcome the immunosuppressive TME. Furthermore, the fact that TiME biomarkers are significant in multivariate analysis of RFS but not OS suggests that immunotherapy might be more effective as an adjuvant treatment for reducing recurrence than for treating recurrent or metastatic disease. This highlights the need to increase the number of CD8^+^ T cells with pre- or postoperative immunotherapy to restore the TiME. Recent studies have shown exceptional responses to neoadjuvant immunotherapy with botensilimab (anti-CTLA-4) and balstilimab (anti-PD-1) for pMMR/MSS colorectal cancer, suggesting another approach to improve tumor regression and restore the TiME [36].

Finally, our study re-evaluated clinical factors currently used within the same stage. Despite its correlation with CD8^+^ T cell expression, the NLR alone is insufficient for prognosis prediction; thus, new blood-based immune biomarkers are needed. Although TRG and NAR are widely used, they have limitations in predicting recurrence in patients with the same disease stage. Therefore, the development of a combined prediction model that incorporates both clinical and TiME factors is warranted.

Our study has several limitations. First, the study had a retrospective design with a small sample size. Thus, we plan to validate the role of AI-based digital pathology as an immune digital biomarker. Further research is required to investigate the correlation between these immune biomarkers and immune phenotypes, particularly through transcriptomic analyses and other advanced techniques. Additionally, since the role of the stroma was not addressed in this study, the co-expression of other CAF-related markers must also be confirmed.

One of the strengths of our study is that it demonstrates the importance of the TiME in identifying high-risk patients, even those with lower-stage disease. We confirmed the prognostic utility for predicting recurrence by incorporating digital AI with CD8^+^ T cell expression. As this approach can also provide stratification for postoperative adjuvant chemotherapy in clinical practice, we plan to conduct further research for validation.

## 5. Conclusions

Our study emphasizes the importance of the TiME in identifying high-risk patients, including those with lower-stage disease. By integrating AI-based analysis with CD8^+^ T cell expression, we confirmed the prognostic utility of these biomarkers in predicting recurrence, thereby providing a basis for stratifying patients for postoperative adjuvant chemotherapy in clinical practice. Further research is needed to validate these findings and explore the role of additional TME components.

## Figures and Tables

**Figure 1 cancers-16-03353-f001:**
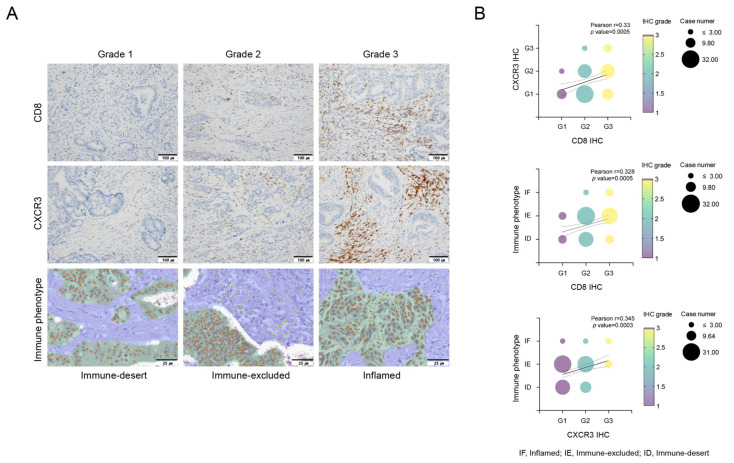
Evaluation of CD8+ T cells, CXCR3 expression in tumor-infiltrating lymphocytes (TILs), and immune phenotypes via Lunit SCOPE IO, and their correlations. Immunohistochemical evaluation of infiltration of CD8^+^ T cells and CXCR3 in tumor-infiltrating lymphocytes (TILs). Grade 1, no or low expression; Grade 2, moderate expression; and Grade 3, high expression. Lunit SCOPE IO infers segmentation of the cancer epithelium (green), cancer stroma (blue), and TIL (yellow dot). The definitions of inflamed, immune-excluded, and immune-desert are described in the Methods section (**A**). The analysis demonstrated that increased CD8^+^ T cell infiltration was positively associated with increased CXCR3 expression in tumor-infiltrating lymphocytes (TILs) and an inflamed immunophenotype, indicating a favorable immune response (**B**). Correlations between CD8^+^ T cells, CXCR3 and immune phenotypes (**B**). IF, inflamed; IE, immune-excluded; ID, immune-desert.

**Figure 2 cancers-16-03353-f002:**
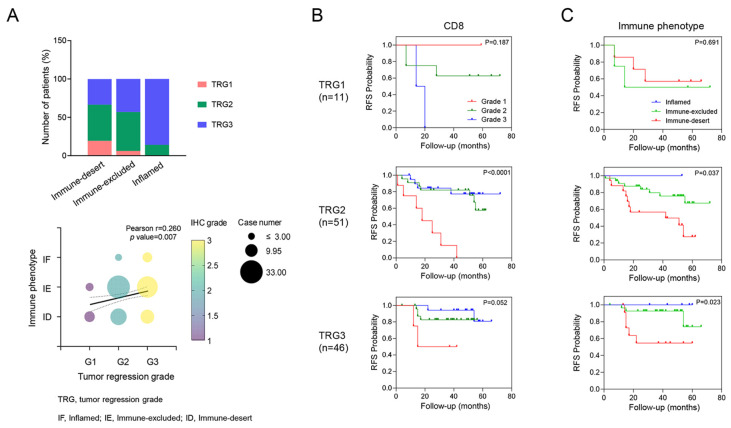
Correlation between tumor regression grade (TRG) and tumor immune microenvironment (TiME) biomarker. A correlation was shown where higher TRG tended to be associated with a more inflamed phenotype (**A**). Kaplan–Meier survival analysis of RFS in each TRG according to the TiME biomarkers, including CD8 and immune phenotypes (**B**,**C**). This panel shows how different levels of TiME biomarkers impact RFS within each TRG category, highlighting their prognostic value.

**Figure 3 cancers-16-03353-f003:**
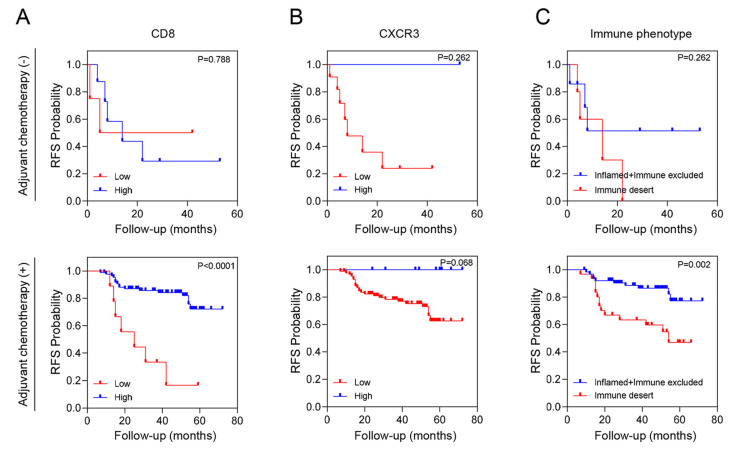
RFS according to tumor immune microenvironment (TiME) biomarkers and adjuvant chemotherapy. (**A**) CD8 expression, (**B**) CXCR3 expression and (**C**) Immune phenotype.

**Figure 4 cancers-16-03353-f004:**
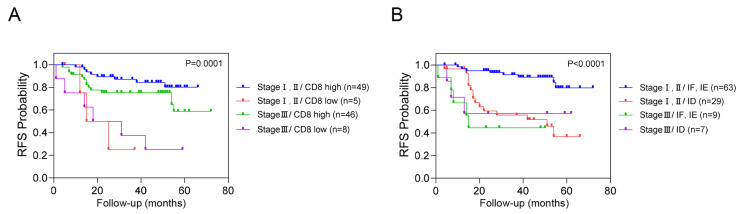
Combined analysis of RFS based on yp stage, CD8^+^ T cells, and immune phenotypes. This figure illustrates the combined analysis of recurrence-free survival (RFS) based on yp stage with CD8+ T cells (**A**) and immune phenotype (**B**).

**Table 1 cancers-16-03353-t001:** Clinical characteristics of the studied patient cohort.

Characteristics	*N* (%)
Sex	
Male	79 (73.1)
Female	29 (26.9)
Age	
Median (year)	60 (36–90)
<70	69
>70	39
Tumor site	
upper-mid	77 (71.2)
low	31 (28.7)
Clinical stage at diagnosis	
T Stage	
T2–3	44 (61.1)
T4	42 (38.9)
N Stage	
N0–1	62 (57.4)
N2	46 (42.6)
Adjuvant chemotherapy	
None	12 (11)
5-FU monotherapy	41 (38)
5-FU + oxalipaltin	55 (51)

5-FU, 5-fluorouracil.

**Table 2 cancers-16-03353-t002:** Univariate and multivariate analyses associated with RFS and OS in patients with stage III LARC after nCRT.

		RFS				OS			
Characteristics	*N*	Univariate Analysis		Multivariate Analysis		Univariate Analysis		Multivariate Analysis	
		HR (95% CI)	*p-*Value	HR (95% CI)	*p-*Value	HR (95% CI)	*p-*Value	HR (95% CI)	*p-*Value
Sex									
Female	29	1				1			
Male	79	1.4 (0.7–3.0)	0.347			1.7 (0.6–5.1)	0.300		
Age									
<70	39	1				1			
≥70	69	1.4 (0.7–3.0)	0.326			1.7 (0.6–4.6)	0.321		
Site									
Upper-mid	77	1				1			
Low	31	2.5 (1.2–5.0)	**0.010**			2.4 (0.9–6.3)	0.077		
Tumor grade									
G 1–2	99	1				1			
G 3	9	4.3 (1.6–11.3)	**0.016**	3.7 (1.3–11.0)	**0.016**	4.4 (1.2–15.6)	**0.012**		
ypT									
ypT 0–3	95	1				1			
ypT 4	13	2.4 (1.0–5.8)	0.051			4.3 (1.5–12.4)	**0.003**		
ypN									
ypN 0–1	97	1				1			
ypN 2	11	2.4 (0.9–6.3)	0.064			1.4 (0.3–6.0)	0.681		
LVI									
No	86	1				1			
Yes	17	2.2 (1.0–4.7)	**0.046**			4.6 (1.6–13.3)	**0.002**	4.8 (1.5–15.5)	**0.008**
PNI									
No	61	1				1			
Yes	47	5.9 (2.5–13.6)	**<0.01**	5.3 (2.2–13.1)	**<0.01**	22.6 (3.0–170.6)	**<0.01**	12.6 (1.6–98.8)	**0.016**
Tumor deposit									
No	91	1				1			
Yes	17	2.1 (1.0–4.6)	**0.050**			4.2 (1.6–11.3)	**0.002**		
MSI									
N/A	13								
Yes	92	1				1			
No	3	1.2 (0.1–13.4)	0.876			21.0 (0–732)	0.249		
CRM status									
>1 mm	82	1				1			
≤1 mm	26	3.9 (2.0–8.0)	**<0.01**	3.1 (1.5–6.7)	**0.004**	6.7 (2.4–18.2)	**<0.01**	3.9 (1.4–11.0)	**0.011**
mDworak TRG									
1–2	62	1				1			
3	46	2.2 (1.0–5.0)	**0.048**			2.1 (0.7–6.4)	0.088		
NAR									
<16	55	1				1			
≥16	33	1.8 (0.9–3.6)	0.164			1.4 (0.5–3.7)	0.512		
NLR									
<6	88	1				1			
≥6	20	2.4 (1.1–5.2)	**0.024**			2.5 (0.9–7.2)	0.080		
Adjuvant chemotherapy									
Yes	96	1				1			
No	12	6.2 (2.6–14.7)	**<0.01**	5.1 (1.9–13.5)	**0.001**	10.6 (3.4–33.2)	**<0.01**	6.9 (1.9–25.3)	**0.003**
CD8+ T cell									
High, 2+, 3+	95	1				1			
Low, 1+	13	4.7 (2.2–10.4)	**<0.01**	2.3 (1.0–5.0)	**0.046**	2.4 (0.8–7.5)	0.128		
CXCR3									
High, 3+	10	1				1			
Low, 1+, 2+	98	27.6 (0.3–249.6)	**0.023**			30.1 (0.7–233)	0.081		
CXCL10									
High, 3+	22	1				1			
Low, 1+, 2+	86	1.1 (0.4–2.6)	0.640			1.3 (0.6–2.7)	0.473		
α-SMA									
High 2+, 3+	74	1.8 (0.8–4.2)	0.055			1.2 (0.4–3.5)			
Low 1+	34	1				1	0.349		
AI-immune phenotype									
Inflamed/Immune-excluded	72	1				1			
Immune-desert	36	3.3 (1.6–6.8)	**<0.01**	2.7 (1.3–5.7)	**0.010**	4.3 (1.5–12.3)	**0.004**		

Bold values indicate statistical significance set at *p* < 0.05; LVI, lymphovascular invasion; PNI, perineural invasion; MSI, microsatellite instability; CRM, circumferential resection margin; mDworak TRG, modified Dworak tumor regression grade; NAR, neoadjuvant rectal; NLR, neutrophil-to-lymphocyte ratio; AI, artificial intelligence; CI, confidence interval; HR, hazard ratio; N/A, not applicable; RFS, recurrence-free survival; OS, overall survival.

## Data Availability

Data that support the results of this study are available from the corresponding author, SH CHO upon reasonable request.

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
