# Peer review of "Tumor Immune Microenvironment Biomarkers for Recurrence Prediction in Locally Advanced Rectal Cancer Patients after Neoadjuvant Chemoradiotherapy"

_cancers, 2024, doi:10.3390/cancers16193353_

Round 1

Reviewer 1 Report

Comments and Suggestions for Authors

The authors proposed to use immunohistochemical staining and incorporated AI-powered digital pathology to guide personalized treatment in LARC patients receiving nCRT. The authors showed significant correlation between CD8+ T cell, CXCR3 expression in TIL and inflamed phenotypes, which are also associated with better recurrence-free survival outcomes. There are a few comments for improvement:

1.        Figure 1: “Grade 1, no or low expression; Grade 2, moderate expression, Grade 3, high expression.”

·      How these IHC were graded and stratified into three? The authors need to explain.

2.        Results 3.3: “High expression (2+, 3+) of CD8+ T cells was observed in 40 patients (37%), whereas low expression (1+) was observed in 13 patients (12%; Supplement Table 1).”

·      “The expression of CD8+ T cells” doesn’t make sense. Do you mean the presence of CD8+ T cells?

·      What the scale/unit of the “High expression (2+, 3+)” or “low expression (1+)”?

3.        Results 3.3: “In the case of CXCR3 in TILs, high expression (3+) was observed in 10 patients (9%), whereas while low expression (1+) was observed in 55 patients (51%).”

·      The authors showed IHC staining with CXCR3 in Figure 1, however, how the expression of CXCR3 within TME or total CXCR3 expression within the block was missed.

·      Were TILs identified by CD8 staining? The authors suggested the densities of TILs were assessed by Lunit SCOPE IO, however, how TILs were identified were missed in the manuscript.

4.        Figure 2: the legend panel C is missed.

5.        It’s possible that the “inflamed phenotype” identified by Lunit SCOPE IO is driven by the CD8 and/or CXCR3 expression. The authors would need to show there is no significant correlation between the expression of CD8, CXCR3 and the “inflamed phenotype” if the authors suggested these three are independent biomarkers for predicting survival outcomes.

Author Response

The authors proposed to use immunohistochemical staining and incorporated AI-powered digital pathology to guide personalized treatment in LARC patients receiving nCRT. The authors showed significant correlation between CD8+ T cell, CXCR3 expression in TIL and inflamed phenotypes, which are also associated with better recurrence-free survival outcomes. There are a few comments for improvement:

  1. Figure 1: “Grade 1, no or low expression; Grade 2, moderate expression, Grade 3, high expression.”
  • How these IHC were graded and stratified into three? The authors need to explain.

      → Thank you for pointing this out. We counted the number of positive cells and graded them according to the level of protein expression, as described in methods section (page 8, line 1-6).

  1. Results 3.3: “High expression (2+, 3+) of CD8+ T cells was observed in 40 patients (37%), whereas low expression (1+) was observed in 13 patients (12%; Supplement Table 1).”

→ I appreciate your careful attention to the missing parts. We correct high expression (2+, 3+) to grade 3+ of CD8+ T cells, and low expression (1+) to grade 1+ CD8+ T cells to clarify further as follow,

Page 11, line 25-26 : Grade 3+ of CD8+ T cells was observed in 40 patients (37%), whereas grade 1+ of CD8+ T cells was observed in 13 patients (12%; Supplement Table 1).

â–ª “The expression of CD8+ T cells” doesn’t make sense. Do you mean the presence  of CD8+ T cells?  What the scale/unit of the “High expression (2+, 3+)” or “low expression (1+)”?

      → Thank you for the comment. It seems there was some confusion regarding the IHC grade (grade 1+, 2+, 3+) and grouping (high expression group or low expression group) based on survival analysis (according to supplementary Fig 2).

The high and low expression groups were arbitrarily designated based on the grades that showed similar survival curves, as seen in the supplementary Fig 2 (ex. For CD8+ T cells, the high expression group included grade 2+ and grade 3+ CD8+ T cells which showed similar RFS, while low expression group included grade 1+ CD8+ T cells)

         To avoid confusion, the grades of the IHC results have been labeled only as grade 1+, 2+, 3+. The terms 'high expression group' and 'low expression group' were used only when grouping was applied, and these have been redefined and revised accordingly (page 11, line 25~ page 12, line 13).

  1. Results 3.3: “In the case of CXCR3 in TILs, high expression (3+) was observed in 10 patients (9%), whereas while low expression (1+) was observed in 55 patients (51%).

â–ª The authors showed IHC staining with CXCR3 in Figure 1, however, how the expression of CXCR3 within TME or total CXCR3 expression within the block was missed.

→ Thank you for pointing this out. Since we analyzed the tissue microarray (TMA) block, most of the core was composed of remnant tumor clusters or cells and surrounding stroma, which thought be a TME and we analyzed CXCR3 in TME. Therefore, we have added this comment in the Method section as follows;

page 8, line 2-3 : The immunoreactivity for CXCR3 in TME was interpreted as 1, ≤33% of positively reactive lymphoid cells; 2, 34%–67% of positive cells; and 3, >67% of positive cells.

â–ª Were TILs identified by CD8 staining? The authors suggested the densities of TILs were assessed by Lunit SCOPE IO, however, how TILs were identified were missed in the manuscript.

→ Thank you for pointing this out. We identified TILs as the lymphocytes located in the stroma surrounding the tumor in H&E and IHC slides. The evaluation of expression was performed manually by an expert pathologist. Therefore, we inserted this mention in the text as follows;

page 8, line 6-8: We identified TILs as the lymphocytes located in the stroma surrounding the tumor in H&E and IHC slides. The evaluation of expression was performed manually by an expert pathologist.

→ Independently, Lunit SCOPE IO, the AI model predicts cancer areas (cancer parenchyma), cancer stroma, and lymphocytes at the pixel level. It then identifies lymphocytes located in both the cancer area and cancer stroma as TILs, which are used to differentiate immune phenotypes. This mention was also inserted into the text (page 9, line 12-14)

  1. Figure 2: the legend panel C is missed

→ Thank you very much for identifying the missing part and we have added the legend in Figure 2 (page 25, line 5).

  1. It’s possible that the “inflamed phenotype” identified by Lunit SCOPE IO is driven by the CD8 and/or CXCR3 expression. The authors would need to show there is no significant correlation between the expression of CD8, CXCR3 and the “inflamed phenotype” if the authors suggested these three are independent biomarkers for predicting survival outcome

→ Thank you for the good comment. CXCL10 is a chemokine that plays a critical role in attracting immune cells, including CD8 T cells, to sites of inflammation or tumors. CXCR3, a receptor expressed on the surface of activated CD8 T cells, binds to CXCL10 and other related chemokines. This interaction facilitates the migration of CD8 T cells to areas where they are needed, such as sites of infection or tumor tissue. Thus, CXCL10 and CXCR3 contribute to the recruitment and localization of CD8 T cells, which can enhance the immune response in various pathological contexts.

As your comment, we inserted this content about the relation of CD8 T cell-CXCL10-CXCR3 in introduction section (page 4, line 28- page 5, line 2, Add reference 7)

In addition, we observed a correlation between the expression of CD8, CXCR3, and the inflamed immune phenotype. However, to comprehensively analyze the mechanisms linking these immune markers and phenotypes, further studies, including transcriptome analysis, are needed. We have added this point to the discussion section. (page 16, line 27-29).

Reviewer 2 Report

Comments and Suggestions for Authors

The aim of the study was to identify prognostic factors by combining clinicopathologic parameters with TME biomarkers in case of locally advanced rectal cancer patients after neoadjuvant chemotherapy. The authors describes their goals, their methods and conclusions very clearly, but sometimes it is hard to find a correlation between the results and the conclusions.

There are a lot of abbreviations in the manuscript, but sometimes they are used before they would be explained. For example `yp stage` is used already in the Simple Summary session, but it`s meaning is only described in the Introduction session (same with TiME). The authors analyzed CD8+ T-cells, CXCR3, CXCL10 and a-SMA, but it would be nice to have the reasons behind their choice, so why exactly these were analyzed.
On Page 4, Figure 1 contains different immunophenotypes, but how these categories are made is not described there, only in Paragraph 2.5 on Page 6, so two pages later. Since Figure 1 is already mentioned: scale bar is missing from the bottom three images and this figure seems to be the same as Figure 4.

On Page 6 it is mentioned that seven patients did not receive adjuvant chemotherapy, but it is not clear to me if these patients were excluded or included in the analysis (the title of the manuscript says `patients after neoadjuvant chemotherapy`).

The authors are referring to Table 1 and Table 2, but these tables are not in the manuscript, only the table legends (at least I do not have them in the pdf I received.)

In several analyses the authors mentioned that a certain percentage of patients had high expression, another percentage showed low expression, but these do not add up to 100%. For example on Page 7, Paragraph 3.3: High expression of CD8+ cells was observed in 40 patients (37%), whereas low expression was observed in 13 patients (12%). (Same with CXCR3 expression.) What was the expression level in the rest of the patients?

The authors cited 36 publications, but only 12 of them are from the last 5 years and to my opinion this could be improved.

I am not sure where this issue is coming from, but on the top of the manuscript the header is ``Cancers 2023, 15...` Please update this.

The goal and the methods of the manuscript, as well as the importance of the study are clear, but the manuscript needs to be improved. For this reason I recommend the paper for publication after minor revision.

Author Response

Reviewer 2.

The aim of the study was to identify prognostic factors by combining clinicopathologic parameters with TME biomarkers in case of locally advanced rectal cancer patients after neoadjuvant chemotherapy. The authors describes their goals, their methods and conclusions very clearly, but sometimes it is hard to find a correlation between the results and the conclusions

  1. â–ªThere are a lot of abbreviations in the manuscript, but sometimes they are used before they would be explained. For example `yp stage` is used already in the Simple Summary session, but it`s meaning is only described in the Introduction session (same with TiME).

→ We agree with this comment. Therefore, we modified yp stage into pathologic stage in Simple Summary because the sentences can be long to explain the abbreviation (page 2, line 10). Also, we have explainded and clarified the abbreviation ‘TiME’ (page 2, line 14).

â–ªThe authors analyzed CD8+ T-cells, CXCR3, CXCL10 and a-SMA, but it would be nice to have the reasons behind their choice, so why exactly these were analyzed.

→ We agree with the reviewer’s comment and we have added that content to the introduction (page 4, line 28~page 5, line 2, Add reference 7).

  1. On Page 4, Figure 1 contains different immunophenotypes, but how these categories are made is not described there, only in Paragraph 2.5 on Page 6, so two pages later. Since Figure 1 is already mentioned: scale bar is missing from the bottom three images and this figure seems to be the same as Figure 4.

→ Thank you for pointing this out. As your comment, we have inserted that the definitions of inflamed, immune-excluded and immune-desert are described in the Methods section in the Figure 1 legend (page 25, line 7-8).

â–ªScale bar is missing from the bottom three images and this figure seems to be the same as Figure 4.

→ Thank you very much for identifying the missing part and we have added the bar in the bottom three images in Figure 1.

→ Fig 1 and Fig 4 are different. We have carefully rechecked them and uploaded the correct versions again

  1. On Page 6 it is mentioned that seven patients did not receive adjuvant chemotherapy, but it is not clear to me if these patients were excluded or included in the analysis (the title of the manuscript says `patients after neoadjuvant chemotherapy`).

→ Thank you for a detail comment. There seems to be some confusion regarding the terminology, so to clarify: neoadjuvant chemotherapy typically refers to preoperative chemotherapy, while adjuvant chemotherapy refers to postoperative chemotherapy. To avoid confusion, we have revised the text by adding 'adjuvant (postoperative) chemotherapy' in the initial section for clarity (page 4, line 15).

  1. The authors are referring to Table 1 and Table 2, but these tables are not in the manuscript, only the table legends (at least I do not have them in the pdf I received.)

→ Thank you for identifying our technical mistakes and we have carefully rechecked them and uploaded the tables again.

  1. In several analyses the authors mentioned that a certain percentage of patients had high expression, another percentage showed low expression, but these do not add up to 100%. For example, on Page 7, Paragraph 3.3: High expression of CD8+ cells was observed in 40 patients (37%), whereas low expression was observed in 13 patients (12%). (Same with CXCR3 expression.) What was the expression level in the rest of the patients?

→ I very much appreciated you for the helpful comment. After reviewing your comment, we re-examined the manuscript and recognized that there might be some confusion regarding the grouping based on survival analysis (high expression group or low expression group) based on the IHC grade (1+, 2+, 3+) and error in the description (% of patients). We revised as follows;

<Page 11, line 25~26>

Grade 3+ of CD8+ T cells was observed in 40 patients (37%), whereas grade 1+ of CD8+ T cells was observed in 13 patients (12%; Supplement Table 1).

         To avoid confusion, the grades of the IHC results have been labeled only as grade 1+, 2+, 3+. The terms 'high expression group' and 'low expression group' were used only when grouping was applied, and these have been redefined and revised accordingly (page 11, line 25~ page 12, line 13).

  1. The authors cited 36 publications, but only 12 of them are from the last 5 years and to my opinion this could be improved.

→ Thank you for the comment. We have updated some references with more recent ones, excluding those that provide historical background, and added reference 7 to the introduction.  

  1. I am not sure where this issue is coming from, but on the top of the manuscript the header is ``Cancers 2023, 15...` Please update this.

     → Thank you for pointing this out. We did not upload this phrase, but we will check it carefully again.
